# Factors Associated with College Students’ Human Papillomavirus (HPV) Vaccination and Preferred Strategies for Catch-Up Vaccine Promotion: A Mixed-Methods Study

**DOI:** 10.3390/vaccines11061124

**Published:** 2023-06-20

**Authors:** Seok Won Jin, Yeonggeul Lee, Sohye Lee, Haeun Jin, Heather M. Brandt

**Affiliations:** 1School of Social Work, The University of Memphis, Memphis, TN 38152, USA; 2Department of Medical Humanities and Social Science, College of Medicine, Yonsei University, Seoul 03722, Republic of Korea; 3Institute of Media Arts, Yonsei University, Seoul 03722, Republic of Korea; 4Social Science Research, University of Seoul, Seoul 02504, Republic of Korea; lynggl@uos.ac.kr; 5Loewenberg College of Nursing, The University of Memphis, Memphis, TN 38152, USA; slee26@memphis.edu; 6Northside Hospital Duluth, Duluth, GA 30096, USA; haeun.jin@northside.com; 7HPV Cancer Prevention Program, St. Jude Children’s Research Hospital, 262 Danny Thomas Place, Memphis, TN 38105, USA; heather.brandt@stjude.org

**Keywords:** college students, human papillomavirus, HPV vaccine, HPV vaccination, mid-south, mixed-methods study

## Abstract

Human papillomavirus (HPV) vaccination protects against six types of cancer—cervical, anal, oropharyngeal, penile, vulvar, and vaginal. In the United States (U.S.), HPV vaccination coverage in college students remains low, especially in the Mid-South region, despite the highest risk of HPV infections and disease burden. However, few studies have assessed HPV vaccination among college students here. This study examined factors associated with HPV vaccination among college students in the Mid-South and explored preferred strategies for promoting vaccination. A mixed-methods design comprising a cross-sectional, self-report online survey and dyadic virtual interviews was conducted. Simple random sampling was performed to recruit a total of 417 undergraduate students aged 18–26 from March to May 2021; convenience sampling was performed to recruit three sex-matched dyads of a total of six (four female and two male) undergraduates from survey respondents who had not completed the HPV vaccine series in May 2021. Binary logistic regression analyses showed HPV vaccine knowledge and perceived barriers to vaccination were factors contributing to coverage for both female and male students, while perceived risks of HPV and vaccine hesitancy were factors only among female students. Findings from the qualitative content analysis identified college students’ perceived barriers to the vaccination at multiple levels and preferred strategies for vaccination promotion, corroborating the findings from the survey study. The findings provide implications that benefit the development of tailored interventions aimed at facilitating catch-up vaccination among college students in the Mid-South region. There is an urgent need for further research and the implementation of effective strategies that address the identified barriers and improve HPV vaccine uptake in this population.

## 1. Introduction

Human papillomavirus (HPV) is the most common sexually transmitted infection in the United States (U.S.), which occurs through intimate skin-to-skin contact [1]. HPV infects almost 85% of sexually active adults during their lifetime [2]. In particular, college-aged individuals who are in their late adolescence and young adulthood are at the highest risk of HPV infections [3,4,5]. HPV infections are subclinical, lack visible symptoms, and typically last no longer than 2 years [6]. Although benign manifestations are usually observed and individuals with HPV rarely develop symptoms, certain types of low-risk HPV (e.g., HPV 6/11) can cause noncancerous genital warts and high-risk HPV (e.g., HPV 16/18) can cause several precancers and cancers [1,6,7,8,9,10,11]. These cancers include cervical, vaginal, and vulvar cancers for women, penile cancer for men, and oropharyngeal and anal cancers for both women and men [7,12]. HPV is estimated to account for more than 37,000 cancers in the U.S. each year, which incurs approximately USD 775 million in direct medical costs within the U.S. healthcare system [13].

The HPV-attributable cancers are mostly preventable with available vaccines [14]. Since 2006, three vaccines of Cervarix, Gardasil, and Gardasil-9 have been approved for use in the U.S. [9]. Among the vaccines, the Gardasil-9 nonavalent vaccine is widely used in the U.S. due to its safety and effectiveness in protection against most high-risk HPV (e.g., HPV 16/18/31/33/45/52/58), preventing more than 90% of the HPV-attributable cancers [15,16]. Research also has shown that the use of the HPV vaccines was successful in lowering the rates of HPV infections in the U.S. [17]. For example, the infection rates of 4-valent HPV (i.e., HPV 6/11/16/18) declined from 11.5% to 4.3% among females aged 14 to 19 years and from 18.5% to 12.1% among females aged 20 to 24 years, respectively, between the years of 2003 and 2006 (prevaccine introduction) and of 2009 and 2012 (postvaccine introduction); low rates of the 4-valent HPV infections also were observed among males in the year of 2013–2014 after the vaccine introduction for males [17]. Currently, the Advisory Committee on Immunization Practices (ACIP) recommends routine HPV vaccination at ages 11 or 12 years, starting at age 9, and catch-up vaccination for individuals who are not fully vaccinated through the age of 26 [18].

### 1.1. HPV Vaccination among College Students

Despite the availability of a safe, effective vaccine [19,20,21,22,23], HPV vaccination coverage remains suboptimal among U.S. college students [24,25,26]. Across all age-based populations in the U.S., college students have reported high rates of HPV prevalence and low rates of HPV vaccination [12]. A recent nationwide survey revealed that only 53% of college students—about 41% of men and 57% of women—were up to date with the recommended HPV vaccination in 2022 [27], which is lower than 62% of adolescents aged 13 through 17 years in 2021 [28] and falls far below the 80% benchmark of the Healthy People 2030 goal [29]. Prior studies also indicated that nearly one out of five (19.1%) college students were unvaccinated against HPV, and 23 million young adults aged 19 to 26 years were unvaccinated against HPV [27,30]. Additionally, research demonstrated that young adults in the U.S. aged 18 to 26 years were less likely than their younger counterparts to be vaccinated against HPV, while 63% and 80% of these young adults reported the initiation of sexual activity at 18 years old and 20 years old, respectively [31]. Moreover, research has pointed out that college is a prime opportunity for catch-up HPV vaccination to protect against HPV infections and prevent HPV-attributable cancers in that college students can make their own health-related decisions independent of parental consent and access health resources on campus through student health centers [32].

### 1.2. Barriers to HPV Vaccination among College Students

The existing literature has indicated that the low rates of HPV vaccination among college students were associated with several multilevel barriers to the vaccination. At the individual level, the barriers included a lack of awareness and knowledge of HPV vaccination [33], HPV-vaccination-related misperceptions or misbeliefs (e.g., no perceived need for vaccination, low perceived susceptibility, the belief of being too late to get vaccinated against HPV in college, and linking the vaccination to the idea of encouragement of sexual activity) [34], and hesitancy for HPV vaccination based on concerns about safety and side effects [3,12]. At the societal level, stigma or norms [3,4,5,9,12,15], parental or partner’s negative attitudes toward or lack of endorsement of the vaccination [5,17,34], or misinformation about HPV and the vaccine (e.g., ineffectiveness, serious side effects, and use only for females) [5,15] could prevent college students from receiving the vaccination. Furthermore, the healthcare-system-level barriers included a lack of access to healthcare providers [4,34], a lack of recommendation for the vaccination [34], costs and insufficient insurance coverage for vaccination [4,17,31], and complex vaccination schedules [31]. Finally, at the structural level, a lack of availability of the vaccine on college campuses and a lack of policies or programs in place to promote vaccination could also be barriers to HPV vaccination among college students [9].

### 1.3. Facilitators to HPV Vaccination among College Students

Research has also shown that several factors could facilitate HPV vaccination among college students. Firstly, education about HPV and the vaccine was positively connected to increasing HPV vaccine intent and uptake among college students [35]. Several intervention studies demonstrated that HPV vaccine education, through a wide range of delivery venues (e.g., videos; social media; and campus-wide campaigns using posters, yard signs, and social media posts), was effective in improving awareness and knowledge of HPV and the vaccine, eventually leading to increased vaccine willingness and uptake among college students [9,17,31]. According to systematic reviews, it is also true that multiple HPV vaccine education interventions among college students increased their knowledge of HPV vaccination, but the increased knowledge did not lead to improving their vaccination outcomes [12,15]. In response to the mixed results, research also provided several suggestions for HPV vaccination education, highlighting that HPV vaccination education should focus not only on improving the awareness and knowledge, but also on the following: (a) increasing perceived susceptibility and severity to HPV and perceived benefits of HPV vaccination and (b) reducing barriers to HPV vaccination and gender disparities in awareness and knowledge and uptake [30,32,35]. Secondly, positive HPV vaccination outcomes were predicted by HPV vaccine recommendation or support from healthcare providers, parents, or peers [31,34]. Especially, strong recommendations by healthcare providers were found to be the most effective strategy for positively influencing the vaccine uptake [5]. Healthcare providers played an important role in communicating with and educating college students to lower stigma about HPV and concerns over the vaccine [3]. Lastly, improving access to the HPV vaccine could help college students receive the vaccine [30,31]. Previous studies have showed that student health centers on campus might be uniquely and optimally positioned to provide catch-up vaccination for college students at free or low cost [17,30,31,35]. According to a nationwide survey, above 70% of 4-year colleges offered the HPV vaccine and more than 95% of college students had health insurance, whereas those having no health insurance should otherwise pay roughly USD 230 per dose for three doses of HPV vaccine [12]. Moreover, a study showed that college health center websites lacked appropriate information about HPV vaccination, including the vaccine availability or costs [30].

### 1.4. Research Gaps

To reach the national goal of HPV vaccination and reduce the burden of HPV-attributable cancers, it is imperative to advance catch-up HPV vaccination in the college-aged population through addressing factors influencing the vaccination. However, public health efforts have shifted primarily to adolescent HPV vaccination, and considerable research has investigated factors contributing to parental decision making for adolescent HPV vaccination [17]. By comparison, few studies have examined the factors contributing to HPV-vaccine-related decision making among college students, although they can take control over their own healthcare [5,31,32]. More specifically, while prior studies indicated that the evident sex disparities in HPV vaccination outcomes existed [15,30,32,35], little is still known about how the factors of the HPV vaccination differ by sex among college students. Additionally, because prior research has reported the mixed results from educational interventions in HPV vaccination outcomes [12,15], further studies are warranted to explore effective educational approaches which would reduce the inconsistency in the vaccination outcomes in this population. Data obtained from the present study may provide essential information for developing interventions for HPV vaccination education and communications targeting college students. Additionally, prior research has extensively documented parental hesitancy for HPV vaccination for adolescents [36,37], yet few studies have investigated hesitancy for HPV vaccination among college students. A clearer understanding of college students’ reasons for the vaccine hesitancy can help public health researchers and educators on campus or at community settings design educational content and messages directed toward alleviating the HPV vaccination hesitancy for college students. Furthermore, the existing literature has revealed that psychosocial variables including perceptions and beliefs pertaining to HPV vaccination were crucial for influencing HPV-vaccine-related decision-making process [3,4,9,12,15,17,31,34]. However, there is the unmet need for comprehensive data regarding these psychosocial variables that contribute to facilitating college students to uptake the HPV vaccine. Finally, while the majority of the prior studies have examined quantitative data, it is also necessary to assess qualitative data that help better unravel college students’ concerns about the vaccination underlying the HPV vaccine hesitancy and their preferred media and approaches to effectively reaching college students and delivering the vaccination information and messages.

### 1.5. Study Aim

This study examined factors associated with HPV vaccination among college students in the Mid-South of the U.S. using a mixed-methods approach. The Mid-South in particular is one of the U.S. regions which has reported high rates of HPV prevalence and cervical cancer mortality and low rates of HPV vaccination [28,38], yet little is known about college students’ HPV vaccination in this region. In the present study, the quantitative survey study focused on investigating college students’ knowledge, beliefs, perceptions, and uptake regarding HPV vaccination. Additionally, the qualitative interview study focused on exploring college students’ perceived barriers and preferred strategies for promoting HPV vaccination. The findings across both sources offered important implications for developing interventional strategies for increasing HPV vaccination among college students and eventually might close the gaps existing in the body of literature on catch-up HPV vaccination in the college-aged population. The research questions of this study included:

Research question 1: What were factors associated with HPV vaccination among college students? If any, how did the factors differ by gender?

Research question 2: What were college students’ perceived barriers to HPV vaccination?

Research question 3: What were college students’ preferred media and strategies for promoting catch-up HPV vaccination?

## 2. Materials and Methods

### 2.1. Study Design and Setting

We employed a mixed-methods, sequential explanatory design to collect quantitative and qualitative data on HPV vaccination among college students at an urban public university in the Mid-South of the U.S. region and then consider these data sources independently, as well as to simultaneously elucidate actionable intervention strategies. This university setting was located in a metropolitan area of the west region of Tennessee, which housed more than 21,000 students, offering baccalaureates through doctoral degrees. According to a recent student enrollment report by the university [39], about 60% of the enrolled students were female, above half were aged 18 to 26 years old, and more than three fourths (76%) were in-state students. Roughly 41% were white, with 34% being Black, 8% being Hispanic, and 5% being Asian. About 31% of degree-seeking undergraduates were of first-generation status, while about 46% of degree-seeking undergraduates were Pell grant recipients.

For the quantitative data, we used a descriptive and cross-sectional online survey and, subsequently, for the qualitative data, we used virtual, in-depth, dyadic interviews among the survey participants. The university institutional review board approved this study (IRB#: PRO-FY2021-297).

### 2.2. Survey

#### 2.2.1. Sampling and Data Collection

We performed simple random sampling to recruit participants through university emails. We obtained a total of 8000 student email addresses (38.1%) which had been randomly selected out of all enrolled students by the university office of institutional research. The inclusion criteria for the survey were students who were aged 18 to 26 years and registered for an undergraduate program at the university at the time of the survey. Students below 18 years old, above 26 years old, or at a postbaccalaureate program were excluded because undergraduates were the target population of the HPV vaccination in this study and the ACIP recommended catch-up vaccination through the age of 26 [40].

Data collection occurred via an online survey tool (i.e., “Qualtrics”) between March and May 2021 [41]. We chose an anonymous, self-administered online survey approach due to a limited number of in-person classes held during the COVID-19 pandemic. We developed a survey questionnaire building on the existing literature in collaboration with the HPV Cancer Prevention Program at the St. Jude Children’s Research Hospital as a community research partner. We distributed to all students in the randomly generated list a recruitment email that included a brief description of the study purpose and eligibility and an electronic link and a QR code to the online survey. Interested individuals who had accessed the online survey via the link or QR code were asked to first complete an electronic consent form and a few questions regarding the study eligibility. Individuals who agreed to participate and were eligible only were directed to the online survey. Participants completing the survey received a USD 10 e-gift card as an honorarium.

#### 2.2.2. Measures

HPV vaccination status. We measured participants’ HPV vaccination status as an outcome variable. We assigned vaccinated status when a participant reported having received at least one shot; otherwise, we assigned unvaccinated status.

HPV vaccination knowledge. To measure knowledge of HPV vaccination, we employed Karki and colleagues’ adapted HPV knowledge scale and HPV vaccine knowledge scale both with options of “true” (=1), “false” (=0), and “do not know” (=0) [42]. The HPV knowledge scale in this study consisted of 17 items of statements that assessed knowledge of HPV-related infection, outcomes, and prevention. Example items included: “Most HPV infections do not clear of their own”, “Certain types of HPV can lead to cervical cancer in women”, and “Condoms are not effective in preventing HPV”. Additionally, the HPV vaccine knowledge scale constituted eight items of statements that assessed knowledge of HPV-vaccination-related dose, effectiveness, and recommendations. Example items included: “It is best to receive HPV shot before being sexually active”, “The HPV vaccine will prevent all causes of HPV-related cancers”, and “The HPV vaccine is offered only to sexually active people”. We summed correct responses only, with higher scores of the scale indicating greater knowledge of HPV vaccination.

HPV vaccination beliefs. We assessed beliefs about HPV vaccination among participants with the 12-item HPV belief scale guided by four constructs of the health belief model (HBM) [42]. The HPV belief scale included: two items of perceived susceptibility, three items of perceived severity, three items of perceived benefits, and four items of perceived barriers, with options on a four-point Likert scale, ranging from “strongly disagree” to “strongly agree”. An example item of perceived susceptibility included “I am at risk of getting HPV”; an example item of perceived severity included “HPV-related cancer is a life threatening disease”; an example item of perceived benefits included “The HPV vaccine will be effective in preventing HPV infection”; and an example item of perceived barriers included “It is hard to find a doctor or clinic that has the vaccine available”. Cronbach’s alpha tests of reliability for this scale showed 0.887 for perceived susceptibility, 0.613 for perceived severity, 0.710 for perceived benefits, and 0.735 for perceived barriers.

Perceived risks of HPV. We measured participants’ perceptions of HPV-related risks using Klasko-Foster and colleagues’ HPV perceived risk scale [30]. Participants were asked to respond to four items of questions regarding the chance to get genital warts and HPV-attributable cancers with options of “no chance” (=0), “low chance” (=1), “moderate chance” (=2), and “high chance” (=3). Example items included “Without the HPV vaccine, what do you think is the chance that you will ever get genital warts?” and “Without the HPV vaccine, what do you think is the chance that you will ever get cervical cancer?” We computed the responses, and higher scores of the scale indicated greater perceived risks of HPV. Cronbach’s alpha of the four items was 0.887.

HPV vaccination hesitancy. To assess participants’ hesitancy for HPV vaccination, we adapted Szilagyi and colleagues’ vaccine hesitancy scale [43]. The hesitancy scale consisted of eight items of statements with options on a four-point Likert scale, ranging from “strongly disagree” (=1) to “strongly agree” (=4). Example items included: “The HPV vaccine is beneficial for me” and “Getting the HPV vaccine is important for the health of others in my community”, and “I am concerned about serious side effects of the HPV vaccine”. After reversely coding positive statements, we computed all scores, with a higher mean score indicating a higher level of HPV vaccination hesitancy. The Cronbach’s alpha of the eight-item scale was 0.875.

Influences on HPV vaccination. We employed a single item to ask participants to report the most important influence on receiving the HPV vaccine by selecting one out of a possible six influences. These influences included: personal views and beliefs; healthcare provider’s recommendation; parent/partner/significant other or mass media (e.g., TV or radio) recommendation; social media (e.g., Twitter, Facebook, or Instagram) recommendation; and accessibility (e.g., ease of obtaining the vaccine and cost of the vaccine).

Preferred cancer-related information sources. To assess participants’ recent cancer-related information source during the COVID-19 pandemic, we asked participants to respond to a question of “The most recent time you looked for cancer information during the COVID-19 pandemic, where did you go first?” The options of this question included: books; brochures, pamphlets, etc.; cancer organization; family; friend/co-worker; doctor or health care provider; internet; social media (e.g., Twitter, Facebook, or Instagram); library; newspapers; telephone information number; and complementary, alternative, or unconventional practitioner. Additionally, we assessed preferred cancer-related information sources when needed in future using a single question with the same option as the above, “Imagine that you had a strong need to get information about cancer during the COVID-19 pandemic. Where would you go first?”

Sociodemographic information. We measured sociodemographic characteristics, including age, gender (female, male, transgender, and prefer not to answer), race/ethnicity, school year, annual household income, religious importance, health insurance, having a primary care provider, and self-report health status.

#### 2.2.3. Data Analysis

We performed descriptive statistics and univariate analyses to examine all variables and HPV vaccination status among participants. We ran bivariate analyses to assess the characteristics of the variables by gender. Additionally, we performed ANOVA with Scheffé test for continuous variables, while we performed a chi-squared test for categorical variables. Finally, we ran binary logistic regression analyses to examine the associations between the independent variables and the HPV vaccination status (i.e., unvaccinated vs. vaccinated) among female and male participants, separately. We employed listwise deletion to handle missing data. We used Stata 14.2 (StataCorp, College Station, TX, USA) for data analysis.

### 2.3. Dyadic Interview

#### 2.3.1. Sampling and Data Collection

For better understandings of HPV vaccination among college students, we performed purposive sampling to recruit interview participants among those who had completed the survey. Those eligible for the interviews were the participants who had completed the survey and were not up to date with the vaccination series. We chose the qualitative approach of a dyadic interview over a focus group or an individual interview as the dyadic interview allowed the researchers to quickly recruit participants and two participants to interact with each other and provide in-depth and detailed information [44]. We also employed a sex-specific dyad to maximize the comfortability of the interviews in which two participants could fully interact in response to questions about even sensitive topics including HPV. Additionally, we used a Zoom-based virtual interview to minimize social contact during the COVID-19 pandemic.

For data collection, we developed a semi-structured interview guide of open-ended questions using the existing literature and informed by the quantitative survey domains. We sent a recruitment email through the university email to potential participants between the last week of April and the first week of May in 2021, but prior to the end of the semester. A total of 18 students (13 females and 5 males) who were interested responded to the email. We contacted the respondents via email to screen the eligibility and obtain their permission to schedule interviews. Among the 18 respondents, a total of 6 respondents (4 female and 2 male students) agreed to participate in an interview, resulting in two female dyads and one male dyad. The first author conducted a total of three dyadic interviews between the second and the third week of May in 2021. At the beginning of each interview, all participants provided verbal consent. During the interview, the interviewer used a funnel approach to ask a broad question followed by more specific questions to probe respondents’ answers. Each interview took about 60 min to complete. We digitally recorded all interviews via Zoom with participants’ permission. The respondents received a USD 20 e-gift card as an honorarium.

#### 2.3.2. Data Analysis

We analyzed the interview data transcribed verbatim in a three-stage process, using inductive coding and qualitative content analysis applying the socioecological model (SEM) as a conceptual framework [45,46]. The SEM allows for identifying multilevel factors that influence individual health outcomes as results from reciprocal complex interactions between the person and the person’s environment [47]. In the current study, the SEM consisted of the three levels of influence—individual (knowledge, beliefs, and perceptions), interpersonal (social network, social support, and patient–provider communications), and community (norms, institutional/organizational factors, programs, and policy) levels—to describe college students’ perceptions toward the barriers and facilitators of HPV vaccination.

In the first stage, the first (S.W.J.), third (S.L.), and fourth authors (H.J.), respectively, read and reviewed all transcripts multiple times to fully understand the responses before coding. As a coder, each author conducted first- and second-level coding independently to generate a joint codebook which included all interview concepts along with corresponding direct quotes. Then, all coders reviewed the codes together as a group and resolved any discrepancies through discussion and consensus. In the second stage, the coders revisited the codebook and transcripts to conduct third-level coding to finalize the codes and categorize the finalized codes into either barriers or facilitators. In the final and third stage, the coders repeated the first-stage process again, categorizing all codes by the barriers and facilitators at the three SEM-based levels, respectively, and identified emerging themes by building relationships among the codes and categories through iteratively comparing the results.

### 2.4. Data Synthesis

Consistent with the mixed-methods sequential explanatory study design, we considered the quantitative data and qualitative data in combination to better understand and identify emergent action steps to improve HPV vaccination coverage. The survey data revealed characteristics or attributes associated with vaccination status (yes/no), which offered us an opportunity to explore how unvaccinated participants may explain their status to inform barriers and facilitators in a more detailed manner. This process included a review of the data and discussion of the implications overall, but specifically among the unvaccinated participants.

## 3. Results

### 3.1. Survey

#### 3.1.1. Sociodemographic Characteristics and Sex Differences of the Sample

Table 1 shows descriptive statistics of sociodemographic characteristics of the participants. Among 8000 potential participants in the sampling frame, a total of 417 students (5.2%) completed the survey. About three quarters (75.5%, n = 312) self-identified as female: 24.2% (n = 101) as male, and less than 1% (n = 4) as transgender or prefer not to answer. Due to the small participant number (n = 4), sex in the analyses was used as a dichotomous (female vs. male) variable. About 77% (n = 319) were 22 years old or younger. Less than half (46.8%, n = 188) were white, while 37.3% (n = 150) and 15.9% (n = 64) were Black and other race or ethnicity, respectively. Roughly 73% (n = 292) were in their third year or above, and about 67% (n = 266) reported an annual household income of USD 40,000 or below. The majority (74.9%, n = 299) reported that religion was important, had a medical health insurance (85.3%, n = 342), and had a medical healthcare provider (73.1%, n = 293). Finally, about 73% (n = 291) self-rated health status as good or excellent. Overall, compared to the characteristics of the student body at the university, the sample over-represented female students (60%, university vs. 76%, sample), while representing a similar distribution of race/ethnicity.

#### 3.1.2. Bivariate Analyses of Variables of Interest for Female and Male Students

Table 2 presents the results from the *t*-test of the variables of interest for female and male participants. Female students had significantly greater knowledge of HPV (t = 2.44, *p* < 0.015) and receipt of the HPV vaccine (t = 3.32, *p* < 0.001) compared to male students. With regard to the HBM-based beliefs of HPV vaccination, male students had significantly higher levels of perceived severity (t = 4.01, *p* < 0.001), perceived barriers (t = 4.54, *p* < 0.001), and perceived susceptibility (t = 2.98, *p* < 0.01) compared to female students. No significant difference was found in the perceived benefits. Moreover, female students had a significantly greater perceived risk for HPV (t = 2.23, *p* < 0.05) compared to male students. Lastly, there was no significant difference in HPV vaccination hesitancy between the two groups.

#### 3.1.3. Distributions of HPV Vaccination Status and Influences on HPV Vaccination for Female and Male Students

Approximately 53% (n = 163) of female students reported having received at least one shot of the HPV vaccine compared to 36% (n = 36) of male students. A *χ^2^* test showed a significant sex difference in the vaccination status (*χ*^2^ = 9.09, *p* < 0.01).

Figure 1 shows the distributions of factors influencing HPV vaccination for female and male students. As the most influential factor on HPV vaccination, female students reported a recommendation from their healthcare provider (42%), followed by personal views and beliefs (36%), a recommendation from a parent, partner, or significant other (12%), accessibility (9%), and mass/social media’s recommendation (4%). In comparison, male students reported a recommendation from their healthcare provider (44%) as the biggest influence on their HPV vaccination, followed by personal views and beliefs (31%), accessibility (13%), a recommendation from a parent, partner, or significant other (9%), and mass/social media recommendation (2%). However, no significant differences in the distribution of factors influencing vaccination were found between female and male students.

#### 3.1.4. Distributions of Cancer-Related Information Sources for Female and Male Students

Participants were also asked to report their preferred sources of cancer-related information during the COVID-19 pandemic. For the sources that female students had recently sought, the Internet (56%, n = 167) was a primary source, followed by other (13%, n = 39), a doctor or healthcare provider (11%, n = 33), social media (7%, n = 21), and family (4%, n = 11). Similarly, for the sources that male students had recently sought during the COVID-19 pandemic, the Internet (57%, n = 56) was a primary source, followed by other (17%, n = 17), a doctor or healthcare provider (71%, n = 7), social media (5%, n = 5), and family (5%, n = 5). Additionally, participants were asked to report their preferred sources of cancer-related information when needed in future. For the sources that female students preferred to seek in future, the Internet (48%, n = 148) was their most preferred source, followed by a doctor or healthcare provider (36%, n = 110), cancer organization (6%, n = 18), and other (4%, n = 11). Likewise, for the sources that male students preferred to seek in future, the Internet (51%, n = 50) was their most preferred source, followed by a doctor or healthcare provider (37%, n = 37), cancer organization (7%, n = 7), and libraries (2%, n = 2).

#### 3.1.5. Binary Logistic Regression for Female and Male Students

Table 3 presents the results from binary logistic regression analyses of HPV vaccination status (unvaccinated vs. vaccinated) for female and male participants. Female students who had greater HPV vaccine knowledge (OR = 1.79, 95% CI [1.38, 2.32]), lower perceived barriers to the vaccination (OR = 0.62, 95% CI [0.49, 0.78]), greater perceived risk of HPV (OR = 1.18, 95% CI [1.04, 1.34]), and lower levels of vaccine hesitancy (OR = 0.86, 95% CI [0.77, 0.97]) were more likely to be vaccinated against HPV, controlling for other variables. Additionally, female students who were older (OR = 1.36, 95% CI [1.03, 1.81]) and more religious (OR = 3.55, 95% CI [1.53, 8.21]) were more likely to be vaccinated against HPV, controlling for other variables. Moreover, male students who had greater HPV vaccine knowledge (OR = 1.57, 95% CI [1.04, 2.39]) and lower perceived barriers to the vaccination (OR = 0.58, 95% CI [0.38, 0.87]) were more likely to be vaccinated against HPV, holding other variables constant. No significant association was found among the control variables for male students.

### 3.2. Dyadic Interview

There were a total of six respondents with four females and two males. Out of the six respondents, one was sophomore, two were junior, and three were seniors across different majors at the university. The analysis and interpretation of the interview data revealed several themes aligning under the following two major topic areas: (a) multilevel barriers to HPV vaccination and (b) preferred strategies for catch-up HPV vaccination promotion.

#### 3.2.1. College Students’ Perceived Barriers to HPV Vaccination

At the individual level, a lack of knowledge of HPV and HPV vaccination emerged as the most common barrier to HPV vaccination among college students. Additionally, the analyses revealed that misperceptions of HPV and the vaccination and concerns and worries about HPV vaccination prevented college students from receiving HPV vaccination at the individual level. Other individual-level barriers included college students’ perceived ignorance of HPV-related information, no priority of the vaccination, and lack of time. At the interpersonal level, a lack of recommendations by healthcare providers, parents, or peers was a barrier to HPV vaccination among college students. The analyses also revealed that parent’s decision was an interpersonal-level barrier to the uptake among college students. Finally, at the community level, social and religious norms were a primary barrier to HPV vaccination for college students. The social and religious norms included stigma of HPV and taboo about discussing the HPV-related topics. These norms generated an atmosphere in which college students were reluctant to openly discuss HPV due to embarrassment and the link to encouragement for sexual activities. Additionally, the structural barriers included difficulties in accessing an HPV vaccine, costs, lack of appropriate information, and absence of education on HPV and the vaccination.

#### 3.2.2. College Students’ Preferred Strategies for Promoting Catch-Up HPV Vaccination

All students mentioned the need for reliable information on HPV vaccination to facilitate college students to receive the vaccines. Students also suggested that the vaccination-related information should include the following content: the benefits and potential risks (e.g., side effects) of getting HPV vaccination, the places (including both campus and local) available for receiving a vaccination, the vaccination guidelines presented by national health institutes (e.g., from the Centers for Disease Control and Protection), and evidence (e.g., on vaccine-related safety and effectiveness) from peer review academic sources. Students also recommended that the information should be designed to target freshmen (e.g., attending orientations on campus); be simple, short, concise, and attractive with visuals (e.g., infographics) for college students to easily and quickly read and understand; and be interpreted into different languages for those who use English as their second language.

Moreover, students suggested several media channels that could effectively deliver information on HPV and the vaccination to college students. These media included university email, printed flyers/brochures, mobile applications, and social media, while students preferred email and social media (i.e., Instagram and Twitter) over others. Particularly, all students highlighted that social media were the most popular platform to effectively reach college students, deliver the information, and communicate on the vaccination among college students. However, students also pointed out some difficulties in using email and mobile applications. When any information in the university email was long and text-condensed, college students tended to skip or ignore the information. Additionally, one student stated that college students might be reluctant to use a mobile application because downloading an application which would rarely be used into their own mobile phone limited the internal storage space of the phone. Another student suggested that one of the university social media platforms (e.g., “Human Confessions” on Twitter), which was widely used to post interesting stories among college students at the university, could be an effective way to reach students at the university and deliver the vaccination-related information to them.

Furthermore, students stated that the provision of appropriate information on HPV and the vaccination via commercials, healthcare providers, parents, or peers could promote college students’ HPV vaccination. Additionally, students mentioned that the recommendations by healthcare providers were the most trusted source of information on HPV and HPV vaccination and a primary facilitator to receiving HPV vaccination for college students. Students also suggested campus-wide initiatives to encourage HPV vaccination, such as a tabling event with giveaways (e.g., T-shirts) and e-gift cards. Multiple students pointed out that the student health center could be an accessible place to obtain information and free vaccination for college students.

Lastly, for the feasibility of virtual reality (VR) in HPV vaccination education for college students, students were asked to describe their thoughts on using VR for educating college students on HPV vaccination. Most students stated that they were familiar with VR because of their past experiences of using it to play games, indicating that these experiences were interesting and fun. Students also revealed positive attitudes toward VR as an unexplored-yet-promising tool to educating college students on HPV vaccination, while the male dyad showed more experiences and stronger interests in using VR for vaccination education. In particular, both male students indicated that VR could attract college students’ attention to vaccination-related information, expressing their willingness to watch the information through VR. Additionally, a female student stated that using VR might be a better approach to promoting HPV vaccination among college students, compared to other ones such as social media and on-campus tabling events, because VR might enable the information to reach college students whom other approaches might miss. However, while most students stated that using VR would be a breakthrough approach for HPV vaccination education for college students, some students also addressed their concerns about possible difficulties in purchasing or accessing a VR device among college students.

## 4. Discussion

Few studies have assessed vaccination hesitancy among college students [48]. One of the objectives of this study was to assess and understand HPV vaccination hesitancy among college students. In this study, contrary to recently published studies on vaccination hesitancy among the U.S. and global populations overall [49], we found little to no vaccination hesitancy among this sample of college students. As with intervention approaches to promote HPV vaccination among college students, taking advantage of the transition period to adulthood and making one’s own decisions are ideal experiences on which to build healthy behaviors [32]. On a related note, respondents recommended targeting incoming first-year students with information about HPV vaccination, including guidelines for routine recommendation, evidence supporting vaccination, benefits and risks, and how to access HPV vaccination. This approach recommended by participants reflects a multilevel orientation to improving HPV vaccination to increase knowledge and address misperceptions, address the influences of a healthcare provider and even parents or caregivers, ensure norms reflect positive associations with HPV vaccination, and ensure access and cost are addressed [50].

Female survey respondents had higher levels of HPV knowledge, higher perceptions of risk of HPV, and were more likely to be vaccinated compared to males. This is consistent with current estimates of HPV vaccination among college students with females about 20 percentage points higher than males overall for HPV vaccination series completion in 2021 [51]. The determinants of HPV vaccination among females and males in this study are also consistent with previously published research [4]. In this study, however, we did find the perceived severity of HPV, perceived barriers to HPV vaccination, and perceived susceptibility to HPV among males to be higher than in females. This could be a product of early educational efforts of HPV vaccination focusing on females only. The U.S. has embraced gender-neutral vaccination approaches for almost 10 years, but first exposure to HPV vaccination for females could only impact these factors among males.

In terms of factors influencing HPV vaccination, minimal differences were noted between males and females. Both males and females cited healthcare provider recommendation as the most important, followed by personal views and beliefs. Preferred information sources about HPV vaccination were the internet for males and females, then healthcare providers, followed by social media. There were minimal differences in preferred primary information sources between during the COVID-19 pandemic and expected information-seeking behavior in the future.

When considering the results of the survey and interviews, several potential areas for intervention were identified. Prior to the interviews, the survey results were reviewed to inform the in-depth interviews. When evaluating and interpreting the results of both forms of data collection in this mixed-methods study, there were several commonalities identified. Respondents and participants noted barriers and facilitators that were highly actionable for this population. Three primary recommendations for interventions emerged.

Results regarding preferred information sources and messaging channels were prescriptive and directive for intervention development and testing. Participants wanted information that was reliable and capitalized on their usual ways of gathering and considering information. In this study, consistent with previous studies, college students indicated support for digital health interventions [52], specifically mentioning email and social media HPV vaccination interventions [12,53,54,55,56]. The autonomy provided for accessing information through these channels aligns with developmental stages for college students. Capitalizing on this period with instructive health education messages to promote positive health behaviors is ideal. Interventions should attend to popular social platforms, such as TikTok, Instagram, and YouTube, in addition to more traditional platforms, such as Facebook and Twitter [12,52].

One specific type of intervention that was explored in the qualitative interviews was VR, which was a familiar concept to participants because of their positive experiences with playing games. A small sample of students responded favorably to this potential approach. Students should carry out this type of intervention on their own to allow them time to revisit the key educational messages shared. The primary limitation was being able to access technology to support participation in a VR-based intervention. VR interventions for college students date back more than 10 years to explorations with influences on alcohol behavior [57,58], physical activity [59,60,61] and healthy eating [62,63], and mental health [64,65,66]. Recently published research has shown VR and augmented reality interventions to be appropriate for building confidence in vaccinations and promoting adherence to vaccination recommendations [67,68,69,70,71]. Real et al. assessed the impact of VR on HPV vaccination in a pilot trial with physicians to include simulations to practice counseling avatars who demonstrated hesitancy [67]. The pilot trial found HPV vaccine initiation increased after exposure to the VR intervention, thus supporting future research [67]. Additional research is needed to understand the widespread acceptability, suitability, and format of VR-based interventions for college students. Parallel evidence with other relevant health behaviors for college students and promising application to vaccinations offers support for the further exploration of VR and other technology-based approaches.

Regardless of intervention type, the importance of linking increased knowledge and confidence with opportunities for action are critical. This college campus has an on-campus health center, but the health center does not offer the HPV vaccine. As noted in the results, college students’ recommended information about HPV vaccination included how to access the HPV vaccine. Among this highly insured population, converting willingness to opportunities for behavioral performance becomes the primary objective of intervention approaches. Fostering linkages between campuses and vaccination administration is an important implementation strategy for future interventions to promote HPV vaccination with college students.

### Limitations

The study has several strengths, including its relatively large quantitative survey sample size for a single college institution, and also the sample’s racial diversity. However, the survey sample was predominantly female, which may limit interpretations related to sex differences between male and females. The survey sample also had a high proportion (85%) of insured respondents with a high level of usual medical care, with almost three-fourths reporting having a health care provider (73%). While the Patient Protection and Affordable Care Act extended benefits of health insurance coverage to dependents up to age 26, we did not assess the nature of coverage, such as parent/caregiver policy or own policy. The college is located in a non-Medicaid expansion state with several noted gaps to preventive health care. However, the sample reported high levels of coverage and access to care. Additionally, although the study employed probability sampling, the sample offered no specific information on the in-state status of the participants. The findings should be carefully interpreted in terms of the characteristics of “Mid-South” college students. The qualitative interview sample was very small in comparison, and the qualitative results should be considered with this in mind. Lastly, the research was conducted during the COVID-19 pandemic. The vaccination attitudes and beliefs reflected in this study may or may not be relevant in the post-pandemic period.

## 5. Conclusions

The results of this study support previously published assertions about the importance of targeting the catch-up population of college students for HPV vaccination. College students are an ideal population with which to intervene to increase coverage and reduce disease burden. Several opportunities for interventions emerged, including addressing multilevel barriers and aligning intervention approaches with preferences and usual ways in which college students access information. Access to HPV vaccination was noted as an important factor when introducing educational interventions. While the sample size was small, college students in this study were favorable towards VR-based interventions to promote HPV vaccination. Future research should test various implementation strategies to address the multilevel barriers identified in this study and related research to understand the impact on HPV vaccination. In addition, innovative intervention approaches, such as those delivered using VR, should be further explored and developed. College students want to learn more about HPV vaccination and how it may help protect them from HPV diseases. It is critical that we act now to provide them with meaningful information.

## Figures and Tables

**Figure 1 vaccines-11-01124-f001:**
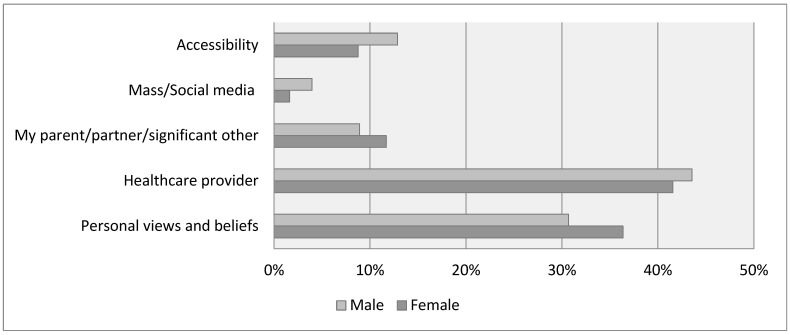
Influences on HPV vaccination.

**Table 1 vaccines-11-01124-t001:** Sample sociodemographic characteristics by gender (N = 417).

Variables	Total (N = 417)		Female (n = 312)		Male (n = 101)		$\chi^{2}$ (*p*-Value)
	n	%	n	%	n	%	
Vaccinated							
Yes	201	48.67	163	52.92	36	35.64	9.09 (0.003)
No	212	51.33	145	47.08	65	64.36	
Age							
20 or below	145	34.77	107	34.29	35	34.65	2.01 (0.570)
21–22	178	42.69	131	41.99	46	45.54	
23–24	65	15.59	49	15.71	16	15.84	
25 or above	29	6.95	25	8.01	4	3.96	
Race							
Black	152	37.44	121	39.93	29	29.29	7.98 (0.018)
White	190	46.80	142	46.86	46	46.46	
Others	64	15.76	40	13.20	24	24.24	
School year							
1st year	32	7.90	21	6.93	10	10.20	4.45 (0.349)
2nd year	80	19.75	54	17.82	24	24.49	
3rd year	124	30.62	93	30.69	30	30.61	
4th year	121	29.88	97	32.01	24	24.49	
5th year	48	11.85	38	12.54	10	10.20	
Annual income							
Less than USD 20,000	175	43.53	135	45.00	39	39.80	3.60 (0.609)
USD 20,001 to USD 40,000	92	22.89	70	23.33	22	22.45	
USD 40,001 to USD 60,000	44	10.95	33	11.00	11	11.22	
USD 60,001 to USD 80,000	29	7.21	19	6.33	9	9.18	
USD 80,001 to USD 100,000	26	6.47	15	5.00	9	9.18	
Over USD 100,000	36	8.96	28	9.33	8	8.16	
Religious importance							
Important	302	74.94	239	79.40	60	61.22	13.01 (0.000)
Not important	101	25.06	62	20.60	38	38.78	
Having medical health insurance							
Yes	435	85.19	260	86.09	82	82.83	0.64 (0.426)
No	60	14.81	42	13.91	17	17.17	
Having a primary health care provider							
Yes	295	72.84	225	74.50	68	68.69	1.28 (0.258)
No	110	27.16	77	25.50	31	31.31	
Health condition							
Very poor/poor/fair	109	27.25	79	26.25	30	30.30	1.95 (0.378)
Good	194	48.50	152	50.50	42	42.42	
Excellent	97	24.25	70	23.26	27	27.27	

Table 1 also shows the results from bivariate analyses of the sociodemographic characteristics by sex. The chi-squared (*χ*^2^) test revealed no significant sex differences in distributions of age, school year, annual household income, medical health insurance, having a primary health care provider, and self-report health status. Furthermore, the chi-squared (*χ*^2^) test showed significant sex differences in distributions of race (*χ*^2^ = 7.98, *p* < 0.05) and religious importance (*χ*^2^ = 13.01, *p* < 0.001).

**Table 2 vaccines-11-01124-t002:** Descriptive analysis of independent variables by gender.

Variables	Total (N = 417)		Female (n = 312)		Male (n = 101)		|t-Value| (*p*-Value)
	Mean	SD	Mean	SD	Mean	SD	
HPV knowledge	3.88	2.41	4.05	2.39	3.35	2.40	2.44 (0.015)
HPV vaccine knowledge	4.82	1.72	4.97	1.63	4.28	1.89	3.32 (0.001)
HPV belief model							
Benefit	9.28	1.54	9.22	1.51	9.48	1.62	1.44 (0.150)
Severity	9.15	1.78	8.95	1.70	9.76	1.90	4.06 (0.000)
Barrier	7.39	2.40	7.10	2.18	8.32	2.80	4.54 (0.000)
Susceptibility	4.11	1.64	3.97	1.57	4.52	1.78	2.98 (0.003)
Perceived risk of HPV	9.40	2.96	9.59	2.96	8.82	3.10	2.23 (0.026)
HPV vaccine hesitancy	14.85	4.41	14.73	4.54	15.24	3.97	1.01 (0.314)

**Table 3 vaccines-11-01124-t003:** Logistic regression analysis of factors associated with vaccination status (unvaccinated vs. vaccinated) by gender.

Variables	Female			Male		
	*OR*	[95% CI]	*p*	*OR*	[95% CI]	*p*
HPV knowledge	1.07	[0.92, 1.25]	0.382	1.07	[0.79, 1.45]	0.678
HPV vaccine knowledge	1.79	[1.38, 2.32]	0.000	1.57	[1.04, 2.39]	0.033
Health belief model						
Benefit	0.98	[0.74, 1.30]	0.877	0.70	[0.39, 1.26]	0.234
Severity	1.00	[0.81, 1.22]	0.970	0.93	[0.61, 1.44]	0.756
Barrier	0.62	[0.49, 0.78]	0.000	0.58	[0.38, 0.87]	0.008
Susceptibility	0.93	[0.72, 1.19]	0.558	0.76	[0.47, 1.22]	0.259
Perceived risk of HPV	1.18	[1.04, 1.34]	0.010	1.04	[0.82, 1.33]	0.726
HPV vaccine hesitancy	0.86	[0.77, 0.97]	0.014	0.86	[0.67, 1.11]	0.254
Age	1.36	[1.03, 1.81]	0.031	0.70	[0.37, 1.33]	0.278
Race (ref. = Black)						
White	0.70	[0.33, 1.50]	0.363	0.39	[0.07, 2.03]	0.260
Others	0.61	[0.20, 1.84]	0.381	0.43	[0.06, 2.93]	0.389
School year	0.67	[0.43, 1.05]	0.079	1.57	[0.63, 3.93]	0.333
Income	0.95	[0.78, 1.16]	0.606	0.80	[0.54, 1.19]	0.265
Religious importance (ref. = Not important)						
Important	3.55	[1.53, 8.21]	0.003	0.78	[0.17, 3.48]	0.742
Insurance (ref. = No)						
Yes	1.14	[0.39, 3.36]	0.813	1.44	[0.20, 10.44]	0.721
Primary healthcare provider (ref. = No)						
Yes	1.33	[0.61, 2.90]	0.475	3.67	[0.69, 19.47]	0.126
Health status	0.95	[0.63, 1.43]	0.807	1.25	[0.43, 3.62]	0.677
Number of observations		281			80	
$\mathrm{LR}\;\chi^{2}$ (*p*)		149.89 (0.000)			39.89 (0.001)	
McFadden’s $R^{2}$		0.389			0.366	

## Data Availability

The data that support the findings of this study are available from the corresponding author upon request.

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
