# Peer review of "Factors Associated with College Students’ Human Papillomavirus (HPV) Vaccination and Preferred Strategies for Catch-Up Vaccine Promotion: A Mixed-Methods Study"

_vaccines, 2023, doi:10.3390/vaccines11061124_

Round 1

Reviewer 1 Report

The paper is interesting and in accord with the literature is quite complete.

Contribution of all authors is significant and could be interesting for scientist.

Analysis and data interpretation are adequacy, writing style could be improving.

I would like the authors to conclude with a proposal to succeed and make more effective information and promotion of the vaccine. This was to improve the clinical impact that the manuscript could have.Moreover, you could add a drawing this it would make reading the manuscript more appealing.

Finally, for a multidisciplinary approach propose to citing:

DOI: 10.1186/s13027-022-00465-9

DOI: 10.3390/jpm12091387 

Author Response

Comments from Reviewer #1:
The paper is interesting and in accord with the literature is quite complete. Contribution of all authors is significant and could be interesting for scientist. Analysis and data interpretation are adequacy, writing style could be improving.
I would like the authors to conclude with a proposal to succeed and make more effective information and promotion of the vaccine. This was to improve the clinical impact that the manuscript could have. Moreover, you could add a drawing this it would make reading the manuscript more appealing.
Finally, for a multidisciplinary approach propose to citing:
DOI: 10.1186/s13027-022-00465-9
DOI: 10.3390/jpm12091387

Authors’ response: We appreciate the reviewer’s comment about recommending a drawing. However, we are unclear for what content we may develop or include a drawing. We noted in the discussion and conclusion about the importance of ensuring college students have information that meets their needs in terms of the clinical aspects of the vaccination. In addition, we also 
have addressed the clinical impact of HPV vaccination, which is a highly effective vaccination when given prior to exposure. We also appreciate the references. We’ve carefully reviewed them and integrated some into the manuscript under 1.4. Research Gaps

Reviewer 2 Report

Review:

The authors have conducted an excellent study and written a great paper on the results. This study outlines the factors influencing college student uptake of the HPV vaccine and conducted in-depth interviews to determine some of the barriers to uptake as well as the preferred ways of promoting vaccination. 

First, a bit of praise:  The introduction was thorough and clear.  The research questions were clearly outlined. The methodology was sound.  The discussion was measured and only made claims that the data supported.

I just have a few suggested improvements, which I will outline below:

1.     76% of the students at this particular college are in-state.  However, it was not stated how many of the sample were in-state.  To characterize this as “Mid-South” students, it would be important to know how many in your actual sample come from the Mid-South.  Are those data available?

2.     I was a little confused about how vaccination status was measured.  In the section 2.2.2, it appeared to be dichotomous and not take into account those who had completed the series, but just grouped together any that had received a shot.  This seems contradictory to what was indicated in 2.3.1 where they indicate that they were looking at those who were “not up to date”, which to me indicated that they had not completed the series.  Some clarity is warranted. 

3.     This is just a suggestion: while Cronbach’s alpha is probably an adequate way to assess the quality of the scale, I might consider running a Confirmatory Factor Analysis on each of the latent variable sin the survey.

4.     There was a small discrepancy in how the “influences on HPV vaccination” variable was reported.  In section 2.2.2, it seemed to indicate that parent/partner/significant other and mass media were together as one variable, and then social media was a different variable. I thought this was odd.  (See lines 274-276).  But, when results were reported, mass/social media were grouped together, as I would have thought they should be. I’m sure this was just an oversight, but it was a little confusing. 

5.     I would like to know more about “religious fidelity” (see line 291).  From what I could see, this was assessed with a single dichotomous variable that indicated whether or not religion was important to a respondent.  This is really kind of a lousy way to address religiosity.  Perhaps we can relabel this as “religious importance” or something?  Later in the paper, the authors indicate that “more religious” females were more likely to vaccinate (see line 443), but this is not what a dichotomous variable like the one you used is measuring. I would not call that a measure of how religious someone is.  And saying “more” would indicate there was a distribution to this variable. But, if it really is dichotomous, there is not a distribution.  They either thought religion was important or they didn’t. Please clarify if this is really how this measure was designed or if I missed something.

6.     I had a hard time differentiating between “perceived susceptibility” and “perceived risk”.  Since it behaved differently between males and females, maybe you could clarify this better. 

7.     Figure 1 seems really unnecessary.  The numbers are listed in the paragraph above and they are really easy to visualize in the reader’s mind.

8.     The discussion of VR in both the results and discussion seems very random and out of place.  I’m not sure why this was included.  Did this spontaneously come up with interviewees or was it prompted? And why? It seems very out of place.  I would consider removing all the VR references unless there is some justification (i.e., literature-backed reason) that we would suspect this method to be successful.

I thoroughly enjoyed reading this paper and I think it has a lot to offer the research community. I look forward to reading a revision.

I found only minor grammatical errors.  Nothing substantial.

Author Response

The authors have conducted an excellent study and written a great paper on the results. This study outlines the factors influencing college student uptake of the HPV vaccine and conducted in-depth interviews to determine some of the barriers to uptake as well as the preferred ways of promoting vaccination. 

First, a bit of praise: The introduction was thorough and clear. The research questions were clearly outlined. The methodology was sound. The discussion was measured and only made claims that the data supported. I just have a few suggested improvements, which I will outline below:
1. 76% of the students at this particular college are in state. However, it was not stated how many of the sample were in-state. To characterize this as “Mid-South” students, it would be important to know how many in your actual sample come from the Mid-South. Are those data available?
Authors’ response: We appreciate this comment. The data we collected provides no specific information of the in-/out-of-state status of the participants. Our recruitment employed a random selection approach, assuming that the sample would represent the characteristics of the study population and the respondents mostly followed the composition of students at the University of Memphis. Only a small proportion of students at the UofM are from out of state (10%). And, the 
vast majority of students are from Tennessee (77%) and then Tennessee, Mississippi, or Arkansas (from these three combined = 86%). Student origins data for UofM are available at https://www.memphis.edu/oir/data/public_student_origin.php. However, we’ve agreed on the importance of the status the reviewer pointed out. Thus, we addressed this in 4.1. Limitations. 

2. I was a little confused about how vaccination status was measured. In the section 2.2.2, it appeared to be dichotomous and not take into account those who had completed the series, but just grouped together any that had received a shot. This seems contradictory to what was indicated in 2.3.1 where they indicate that they were looking at those who were “not up to date”, which to me indicated that they had not completed the series. Some clarity is warranted.

Authors’ response: This study employed a mixed-methods study design. We first conducted quantitative cross-sectional survey and subsequently qualitative interviews. The preliminary analysis of the survey data revealed that among those who had reported the receipt of at least one shot (n = 201), the majority (about 85%) were up to date with the vaccination. Additionally, this study was initially designed to inform development of future interventions for on-campus HPV 
vaccination promotion targeted to the vaccine completion. Thus, we agreed that the interview study focused on better and thoroughly understanding the barriers and facilitators to the vaccination specifically among the participants who had completed the survey and were not up to date with the series.

3. This is just a suggestion: while Cronbach’s alpha is probably an adequate way to assess the quality of the scale, I might consider running a Confirmatory Factor Analysis on each of the latent variable sin the survey.

Authors’ response: If we employed structural equation modeling (SEM) as a primary method, we might run a CFA with latent variables. However, since we employed binary regression and did not include a latent variable in the model, we believe that CFA is not suitable for the analysis process in this study.

4. There was a small discrepancy in how the “influences on HPV vaccination” variable was reported. In section 2.2.2, it seemed to indicate that parent/partner/significant other and mass media were together as one variable, and then social media was a different variable. I thought this was odd. (See lines 274-276). But, when results were reported, mass/social media were grouped together, as I would have thought they should be. I’m sure this was just an oversight, but 
it was a little confusing.

Authors’ response: Because the frequencies of the two items were lower than five, we had to combine them into one. When the frequency of a category is below five, it is not suitable to conduct a chi-squared test.
5. I would like to know more about “religious fidelity” (see line 291). From what I could see, this was assessed with a single dichotomous variable that indicated whether or not religion was important to a respondent. This is really kind of a lousy way to address religiosity. Perhaps we can relabel this as “religious importance” or something? Later in the paper, the authors indicate that “more religious” females were more likely to vaccinate (see line 443), but this is not what a dichotomous variable like the one you used is measuring. I would not call that a measure of how religious someone is. And saying “more” would indicate there was a distribution to this variable. But, if it really is dichotomous, there is not a distribution. They either thought religion was important or they didn’t. Please clarify if this is really how this measure was designed or if I missed something.

Authors’ response: We appreciate this comment. We measured the “religious importance” of the participants using an ordinal variable on a five-point Likert scale with options of: “Extremely important, Very important, Moderately important, Slightly important, and Not at all important.” For the study’s interest in investigating gender-specific factors associated with HPV vaccination and statistical purposes (a chi-squared test and binary logistic regression), we dichotomized the variable into ‘important’ and ‘not important.’ According to the reviewer’s comment, we have relabeled “religious fidelity” as “religious importance” throughout the manuscript. 

6. I had a hard time differentiating between “perceived susceptibility” and “perceived risk”. Since it behaved differently between males and females, maybe you could clarify this better.

Authors’ response: The "perceived susceptibility" as one of the sub-constructs of the Health Belief Model refers to the belief about personal vulnerability to HPV infection, while the "perceived risk" relates to the perception of the potential negative consequences of an HPV infection. As shown in the logistic model of this study, the perception of the potential negative consequences (e.g., cervical cancer) of an HPV infection appear to be a significant factor contributing to HPV vaccination among female students, while it does not among male ones.

7. Figure 1 seems really unnecessary. The numbers are listed in the paragraph above and they are really easy to visualize in the reader’s mind.

Authors’ response: We appreciate the comment. We’ve found the error. According to the suggestion, we’ve removed Figure 1.

8. The discussion of VR in both the results and discussion seems very random and out of place. I’m not sure why this was included. Did this spontaneously come up with interviewees or was it prompted? And why? It seems very out of place. I would consider removing all the VR references unless there is some justification (i.e., literature-backed reason) that we would suspect this method to be successful. 

Authors’ response: We appreciate this comment. In the interviews, we explored respondents’thoughts about using a VR platform as a way of strategies for HPV vaccination promotion that aim to educate HPV and promote HPV vaccination for college students. We included the findings since we agreed that they aligned with an intervention delivery method that may appeal to college students.

I thoroughly enjoyed reading this paper and I think it has a lot to offer the research community. I look forward to reading a revision.

Reviewer 3 Report

Dear Authors,

This paper addresses an interesting topic, however, I would recommend several modifications before considering its publication. Below these are some suggestions for You:

1. Affiliations 

A. Lines 6-19: It should be written according the the Instruction for Authors 

2. Abstract:

A. Line 6: What types of cancer do you mean? 

B. I suggest including more overall informations, which will interest the readers

3. Introduction:

A. The introduction is very informative, maybe too much

B. I recommend including dates regarding clinical as well as molecular aspects of HPV infections like: 1. mainly HPV infections are subclinical and last no longer than 2 years; 2. usually benign manifestations are seen; 3. HPV can evade the innate immune system, delaying the adaptive immune response; infected basal cells during turnover are pushed out towards the epithelial surface, avoiding the circulating immune system, which can promote a persistent HPV infection; 4. It has been established that persistent infection with HPV is associated with cervical, anogenital, as well as head and neck cancers, 5. one of the key events of HPV-induced carcinogenesis is the integration of the HPV genome into a host chromosome; and similar (https://doi.org/10.3390/vaccines10010053, https://doi.org/10.5114/ada.2021.107269, https://doi.org/10.3390/v13040559, https://doi.org/10.3390/v15030778)

C. The aim of the study is well worded

4. Materials and methods: 

A. No fundamental fault was found

B. Line 343: I strongly recommend including the authors’ acronyms 

5. Results:

A. One of the major concerns of the results section is Table 1, and Table 3, which lack transparency, I mean that subgroups/variables should be separated and corrected when needed (eg. plenty of ‘“ref. =” should be removed)

B. In Table 1 there are some flaws (e.g. in the first line, total number of variable ‘vaccinated’ of entire population as well as female are not result of answers within that group, i.e. 212+201=413, not 417 - so do you include or exclude transgender in total population?)

C. Table 2: I suggest checking all data

D. Figure 1: The results should be like in Table 1 variable “vaccinated’, but they are the complete opposite. 

6. Conclusion

A. The conclusions are vague, which will not inspire future research and do not give practical tips. You should underline the most relevant findings (according to the study aim especially) and the values of your study (methodology or results) and include future research possibilities. 

Best regards and good luck

Author Response

Dear Authors,
This paper addresses an interesting topic, however, I would recommend several modifications before considering its publication. Below these are some suggestions for you:
1. Affiliations
A. Lines 6-19: It should be written according to the Instruction for Authors

Authors’ response: We thank this comment. We’ve revised this information, complying with the Instruction for Authors.

2. Abstract:
A. Line 6: What types of cancer do you mean?

Authors’ response: The six types of cancer are added.

B. I suggest including more overall informations, which will interest the readers

Authors’ response: According to the suggestion, we’ve added more overall information to Abstract to interest the readers.

3. Introduction:
A. The introduction is very informative, maybe too much
B. I recommend including dates regarding clinical as well as molecular aspects of HPV infections like: 
1. mainly HPV infections are subclinical and last no longer than 2 years; 
2. usually benign manifestations are seen; 
3. HPV can evade the innate immune system, delaying the adaptive immune response; infected basal cells during turnover are pushed out towards the epithelial surface, avoiding the circulating immune system, which can promote a persistent HPV infection; 
4. It has been established that persistent infection with HPV is associated with cervical, anogenital, as well as head and neck cancers, 
5. one of the key events of HPV-induced carcinogenesis is the integration of the HPV 
genome into a host chromosome; and similar
(https://doi.org/10.3390/vaccines10010053,
https://doi.org/10.5114/ada.2021.107269,
https://doi.org/10.3390/v13040559,

Authors’ response: We appreciate the recommendation along with detailed guidance. We’ve carefully reviewed the sources and then integrated some of them into the manuscript under Introduction.

C. The aim of the study is well worded
4. Materials and methods:
A. No fundamental fault was found
B. Line 343: I strongly recommend including the authors’ acronyms

Authors’ response: According to the recommendation, the authors’ acronyms are inserted.

5. Results:
A. One of the major concerns of the results section is Table 1, and Table 3, which lack transparency, I mean that subgroups/variables should be separated and corrected when needed (eg. plenty of ‘“ref. =” should be removed)

Authors’ response: We appreciate the comment. We’ve carefully reviewed the comment regarding the transparency and formatting of Tables 1 and 3, specifically regarding the separation and correction of subgroups/variables and the removal of unnecessary "ref. =" annotations. Since the variables in Table 1 were categorical, it was necessary to establish a reference group for each of the categorical variable in order to analyze them in the model.

B. In Table 1 there are some flaws (e.g. in the first line, total number of variable ‘vaccinated’ of entire population as well as female are not result of answers within that group, i.e. 212+201=413, not 417 - so do you include or exclude transgender in total population?)

Authors’ response: After double-checking our data analyses and the results thoroughly, we’ve corrected the flaws in Table 1. However, the discrepancy occurred due to missing values for the variable 'sex.' Four participants did not respond to the item regarding their sex.

C. Table 2: I suggest checking all data

Authors’ response: After responding to the comments of A and B above, according to the suggestion, we’ve double-checked our data analyses and the results thoroughly. To our best knowledge, they are in the right shape. 

D. Figure 1: The results should be like in Table 1 variable “vaccinated’, but they are the complete opposite.

Authors’ response: We appreciate the comment. We’ve found the error. According to the comment and the suggestion by another reviewer, Figure 1 is removed.

6. Conclusion

A. The conclusions are vague, which will not inspire future research and do not give practical tips. You should underline the most relevant findings (according to the study aim especially) and the values of your study (methodology or results) and include future research possibilities.

Authors’ response: Thank you for sharing this observation with us! We understand, after further review and reflection, that we could be more prescriptive in our conclusion. Based on the results of our study, we feel there are at least two areas prime for further research. The first is focused on implementation strategies to address multi-level barriers. The second is related to VR-based interventions. We have made edits to offer these suggestions for future research.

Round 2

Reviewer 3 Report

Dear Authors,

This is a revised version of your original manuscript. You have corrected it according to the suggestions and it can be accepted without any changes now.

Best regards and good luck